# DOES ENHANCED SHAPE BIAS IMPROVE NEURAL NETWORK ROBUSTNESS TO COMMON CORRUPTIONS?

**Chaithanya Kumar Mummadi** [*]
University of Freiburg
Bosch Center for Artificial Intelligence
ChaithanyaKumar.Mummadi@bosch.com

**Ranjitha Subramaniam** [*]
Department of Computer Science
TU Chemnitz
ranjivishnu08@gmail.com

**Robin Hutmacher**
Bosch Center for Artificial Intelligence
Robin.Hutmacher@de.bosch.com

**Julien Vitay**
Department of Computer Science
TU Chemnitz
julien.vitay@informatik.tu-chemnitz.de

**Volker Fischer**
Bosch Center for Artificial Intelligence
Volker.Fischer@de.bosch.com

**Jan Hendrik Metzen**
Bosch Center for Artificial Intelligence
JanHendrik.Metzen@de.bosch.com

## ABSTRACT

Convolutional neural networks (CNNs) learn to extract representations of complex features, such as object shapes and textures to solve image recognition tasks. Recent work indicates that CNNs trained on ImageNet are biased towards features that encode textures and that these alone are sufficient to generalize to unseen test data from the same distribution as the training data but often fail to generalize to out-of-distribution data. It has been shown that augmenting the training data with different image styles decreases this texture bias in favor of increased shape bias while at the same time improving robustness to common corruptions, such as noise and blur. Commonly, this is interpreted as shape bias increasing corruption robustness. However, this relationship is only hypothesized. We perform a systematic study of different ways of composing inputs based on natural images, explicit edge information, and stylization. While stylization is essential for achieving high corruption robustness, we do not find a clear correlation between shape bias and robustness. We conclude that the data augmentation caused by style-variation accounts for the improved corruption robustness and increased shape bias is only a byproduct.

## 1 INTRODUCTION

As deep learning is increasingly applied to open-world perception problems in safety-critical domains such as robotics and autonomous driving, its robustness properties become of paramount importance. Generally, a lack of robustness against adversarial examples has been observed (Szegedy et al., 2014; Goodfellow et al., 2015), making physical-world adversarial attacks on perception systems feasible (Kurakin et al., 2017; Eykholt et al., 2018; Lee & Kolter, 2019). In this work, we focus on a different kind of robustness: namely, robustness against naturally occurring common image corruptions. Robustness of image classifiers against such corruptions can be evaluated using the ImageNet-C benchmark (Hendrycks & Dietterich, 2019), in which corruptions such as noise, blur, weather effects, and digital image transformations are simulated. Hendrycks & Dietterich (2019) observed that recent advances in neural architectures increased performance on undistorted data without significant increase in relative corruption robustness.

One hypothesis for the lack of robustness is an over-reliance on non-robust features that generalize well within the distribution used for training but fail to generalize to out-of-distribution data. Ilyas

---

[*]Equal contribution.

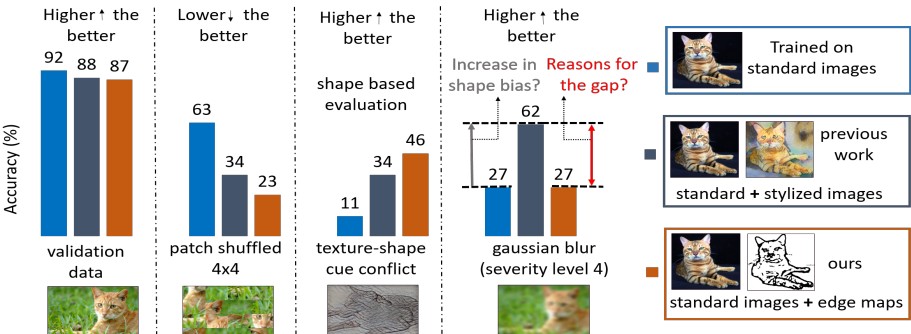

Figure 1: Illustration of the effect of different training augmentations. While both style-based (Geirhos et al., 2019) and edge-based augmentation (this paper) reach the same validation accuracy, edge-based augmentation shows a stronger increase in shape bias as evidenced by lower accuracy on patch-shuffled images and higher rate of classifying according to the shape category for texture-shape cue conflicts. Nevertheless, only style-based augmentation shows a considerable improvement against common corruptions such as Gaussian blur. This challenges the hypothesis that increased shape bias causes improved robustness to corruption.

et al. (2019) provide evidence for this hypothesis on adversarial examples. Similarly, it has been hypothesized that models which rely strongly on texture information are more vulnerable to common corruptions than models based on features encoding shape information (Geirhos et al., 2019; Hendrycks & Dietterich, 2019). Alternative methods for increasing corruption robustness not motivated by enhancing shape bias use more (potentially unlabeled) training data (Xie et al., 2019) or use stronger data augmentation (Lopes et al., 2019; Hendrycks* et al., 2020). Note that our meaning of "shape" & "texture" is built on the definitions by Geirhos et al. (2019).

In this paper, we re-examine the question of whether increasing the shape bias of a model actually helps in terms of corruption robustness. While prior work has found that there are training methods that increase both shape bias and corruption robustness (Geirhos et al., 2019; Hendrycks & Dietterich, 2019), this only establishes a correlation and not a causal relationship. To increase the shape bias, Geirhos et al. (2019) "stylize" images by imposing the style of a painting onto the image, leaving the shape-related structure of the image mostly unchanged while modifying texture cues so that they get largely uninformative of the class. Note that image stylization can be interpreted as a specific form of data augmentation, providing an alternative hypothesis for increased corruption robustness which would leave the enhanced shape bias as a mostly unrelated byproduct.

In this work, we investigate the role of the shape bias for corruption robustness in more detail. We propose two novel methods for increasing the shape bias:

- Similar to Geirhos et al. (2019), we pre-train the CNN on an auxiliary dataset which encourages learning shape features. In contrast to Geirhos et al. (2019) that use stylized images, this dataset consists of the *edge maps* for the training images that are generated using the pre-trained neural network of Liu et al. (2017) for edge detection. This method maintains global object shapes but removes texture-related information, thereby encouraging learning shape-based representations.
- In addition to pre-training on edge maps, we also propose *style randomization* to further enhance the shape bias. Style randomization is based upon sampling parameters of the affine transformations of normalization layers for each input from a uniform distribution.

Our key finding is summarized in Figure 1. While pre-training on stylized images increases both shape bias and corruption robustness, these two quantities are not necessarily correlated: pre-training on edge maps increases the shape bias without consistently helping in terms of corruption robustness. In order to explain this finding, we conduct a systematic study in which we create inputs based on natural images, explicit edge information, and different ways of stylization (see Figure 2 for an illustration). We find that the shape bias gets maximized when combining edge information with stylization without including any texture information (Stylized Edges). However, for maximal corruption robustness, superimposing the image (and thus its textures) on these stylized edges is required. This, however, strongly reduces shape bias. In summary, corruption robustness seems to benefit most from style variation in the vicinity of the image manifold, while shape bias is mostly

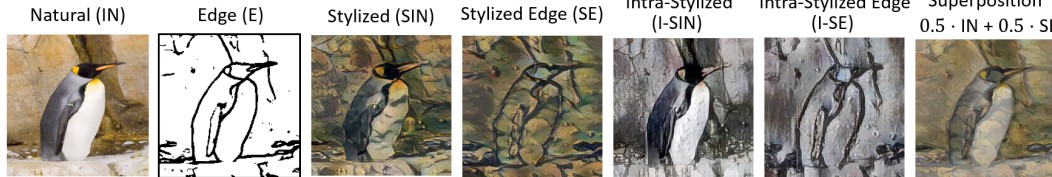

Figure 2: Overview of content and stylization variants used in this paper: Content is a natural image (IN) or an edge map (E). Content is stylized in three different ways: No stylization, style transfer with `Painter by Numbers` as style source as proposed in Geirhos et al. (2019) (SIN and SE), style transfer with a different in-distribution image as style source (I-SIN and I-SE). Additionally, we show a superposition (SE+IN) between natural (IN) and stylized edge image (SE).

unrelated. Thus, image stylization is best interpreted as a strong data augmentation technique that encourages robust representations, regardless whether these representations are shape-based or not.

Moreover, we present results for a setting where we fine-tune only parameters of the affine transformation of a normalization layer on the target distribution (stylized or corrupted images, respectively) for a CNN trained on regular images. Surprisingly, this is already sufficient for increasing the shape bias/corruption robustness considerably. We conclude that CNNs trained on normal images do learn shape-based features and features robust to corruptions but assign little weight to them. It may thus be sufficient to perform augmentation in feature space (extending Nam et al. (2019); Li et al. (2020)) so that higher weights are assigned to features that are robust to relevant domain shifts.

## 2 RELATED WORK

**Texture-vs-Shape Bias** Geirhos et al. (2019) and Baker et al. (2018) hypothesized that CNNs tend to be biased towards textural cues rather than shape cues. This line of research is further supported by Brendel & Bethge (2019), where the authors show that BagNets, Deep Neural Networks (DNN) trained and evaluated only on small restricted local image patches, already perform reasonably well on ImageNet. Similarly, Yin et al. (2019) and Jo & Bengio (2017) showed using a Fourier space analysis that DNNs rely on surface statistical regularities and high-frequency components. The texture-vs-shape bias can be quantified by evaluating a network either on images with texture-shape cue conflict (Geirhos et al., 2019) or on images which were patch-wise shuffled (Luo et al., 2019).

**Robustness Against Common Corruptions** Common corruptions are potentially stochastic image transformations motivated by real-world effects that can be used for evaluating model robustness. Hendrycks & Dietterich (2019) proposed the ImageNet-C dataset that contains simulated corruptions such as noise, blur, weather effects and digital image transformations. Geirhos et al. (2018) showed that humans are more robust to image corruptions than CNNs.

Approaches to improve corruption robustness include data augmentation (Lopes et al., 2019; Yun et al., 2019; Hendrycks* et al., 2020; Cubuk et al., 2019), self-training with more training data (Xie et al., 2019), novel architectures and building blocks (Zhang, 2019; Hu et al., 2018), and changes in the training procedure (Hendrycks et al., 2019; Rusak et al., 2020; Wang et al., 2019). Motivated by the texture-vs-shape hypothesis, Geirhos et al. (2019) and Michaelis et al. (2019) train their network on a stylized version of ImageNet. The idea is that style transfer removes textural cues and models trained on stylized data thus have to rely more on shape information. The observed increase in corruption robustness on this stylized data was attributed to the shape bias. In this work, we provide evidence that contradicts this claim.

Similar to training on stylized images, *Style Blending* (Nam et al., 2019) employs style transfer in latent space by interpolating between feature statistics of different samples in a batch. Li et al. (2020) extend this idea and use feature space blending along with label interpolation. Hendrycks et al. (2019) considers self-supervised training with the prediction of image rotations as an auxiliary task. The authors argue that predicting rotation requires shape information and thus improves robustness. Similarly, Shi et al. (2020) proposes Dropout-like algorithm to reduce the texture bias and thereby increase the shape bias to improve model robustness. However, the authors also discuss that a "sweet

spot" between shape and texture is needed for the model to be robust for domain generalization. With *Patch-wise Adversarial Regularization*, Wang et al. (2019) try to penalize reliance on local predictive representations in early layers and encourage the network to learn global concepts. Other augmentation techniques that aim to improve common corruption robustness are *PatchGaussian* (Lopes et al., 2019), *CutMix* (Yun et al., 2019), *AugMix* (Hendrycks* et al., 2020), and *RandAugment* (Cubuk et al., 2019). At this point, it remains unclear whether the increase in robustness caused by these augmentations is due to learning fundamentally different representations such as more shape-biased ones or to more incremental improvements in feature quality.

**Edge-based Representations** A classical method for extracting edge maps is the Canny edge extractor Canny (1986). More recent approaches use DNNs (Xie & Tu, 2015; Liu et al., 2017) (see Figure A1). Geirhos et al. (2019) evaluate their shape-biased models on edge maps obtained with a Canny edge detector. ImageNet-Sketch (Wang et al., 2019) is a newly collected sketch-like dataset matching the ImageNet validation dataset in shape and size. It is used to evaluate generalization to domain shifts. In contrast to these works, we generate the edge-based representations with an edge detector using Richer Convolutional Features (RCF) (Liu et al., 2017) (see Figure A1) and use them explicitly for training. We provide evidence that edge-based representations enhance the shape bias, through an evaluation on images with induced texture-shape cue conflict and patch-shuffled images.

## 3 LEARNING SHAPE-BASED REPRESENTATIONS

Similar to Geirhos et al. (2019), we aim to enhance the shape bias of a network so that it bases its decision more on shape details than on the style of objects encoded in textures. While Geirhos et al. (2019) augment training data with different styles (stylization), thereby making texture cues less predictive, we extract edge information (edge maps) from the training images to maintain explicit shape details and remove texture-related information completely. Here, we consider grayscale intensity edge maps rather than separate edge maps for each color channel. We propose to train CNNs using the edge maps in addition to the standard training data to learn shape-based representations for more effective shape-based decision-making.

Besides training on the dataset with explicit shape cues, high capacity networks learn different feature representations when trained jointly on datasets from different distributions. Despite edge maps encouraging CNNs to learn shape-based representations, we observe that the network learns to encode features with texture details when introduced to the standard image data during training. We propose here to further restrain the network from learning texture details on standard image data. We discuss below the extraction of edge details from images to create the edge map dataset and explain the technique to reduce the texture bias of the CNN.

**Edge dataset** Given a standard image dataset, we construct a new dataset with edge maps (named the Edge dataset) by extracting the edge details of each image. The edge details are extracted by the CNN-based edge detector using richer convolutional features (RCF) proposed in Liu et al. (2017). RCF network produces a single-channel edge map that contains the pixel values between $[0, 255]$. We convert the non-binary edge map into a binary map with values in $\{0, 255\}$ using a threshold of 128 and transform it into a 3-channel RGB edge map by duplicating the channels, so we can use the edge maps as a direct input to train the CNNs. The edge maps from the Edge dataset are used as input and can be independently used to train or evaluate CNNs without necessarily being combined with the standard image data. Please refer to Section A.1 for the details of RCF network.

**Style Randomization (*SR*)** While using a dataset with explicit shape cues enhances shape-based representations, we propose to further reduce the texture bias of the network when training on standard images. It is shown in the literature of style transfer (Dumoulin et al., 2016; Huang & Belongie, 2017) that the statistics of feature maps (e.g., mean and standard deviation) of a CNN effectively capture the style of an image and changing these statistics would correspond to a change in the style of an image. SIN dataset is generated using such style transfer technique and shown to reduce the texture bias of the networks. Inspired by this observation, we propose a simple technique to effectively reduce the texture bias using the feature statistics when being trained on standard training data. We modify the style of an image in the feature space so that the network becomes style-invariant. In particular, we randomize the style details, i.e. feature statistics, of an image during training such that the network can not rely on the texture cues. A similar approach named *Style Blending* (SB) is proposed in Nam et al. (2019) which randomizes the style information by interpolating the feature

statistics between different samples in a mini-batch. We propose here a slightly different approach to make the network invariant to style information. Instead of interpolating the statistics of similar distribution of data i.e, training samples, we completely randomize the feature statistics (mean and standard deviation) by randomly sampling them from an uniform distribution. Considering $X_i$ as the $i^{th}$ feature map of an intermediate layer in CNN, and $\mu_i$ & $\sigma_i$ as the feature statistics of $X_i$, the style randomized feature map $\hat{X}_i$ is defined as:

$$\hat{X}_i := \hat{\sigma}_i * \left( \frac{X_i - \mu_i}{\sigma_i} \right) + \hat{\mu}_i \tag{1}$$

where $\hat{\sigma}_i \sim \text{Uniform}(0.1, 1)$ and $\hat{\mu}_i \sim \text{Uniform}(-1, 1)$. These specific choices of sampling for $\hat{\sigma}_i$ and $\hat{\mu}_i$ were found to perform best on our evaluations. The style transfer technique described in Huang & Belongie (2017) replaces the feature statistics of content image with the statistics of a desired style image to change the style. Similarly, we replace the statistics of content image with random statistics to change the style information. Training the network with *SR* reduces the texture bias and improves shape-based decision making. An advantage of *SR* over *SB* is that the feature statistics are sampled from a different distribution than the training data, that encourages learning representations to generalize better to out-of-distribution data. We show in Section 5 that *SR* outperforms *SB* and aids the network to induce stronger shape-based representations.

## 4   EXPERIMENTAL SETTINGS

**Dataset** We use a subset of 20 classes from ImageNet dataset (ImageNet20, or IN) that are chosen randomly, to study the role of shape bias towards corruption robustness; the main reason being that extensive experiments on this dataset are feasible with limited computation. Details about this dataset can be found in Section A.2. The Edge dataset of IN (referred to as E) is generated as described in Section 3.

**Stylization variants** In addition to enhancing the shape bias using the edge maps, we further study the contribution of different factors of Stylized ImageNet (SIN) (Geirhos et al., 2019) to gain insights on its improved performance on corruptions. We break down SIN into different factors to understand their influence on corruption robustness. We segregate the factors that jointly generate the stylized images and the factors that are hypothesized to improve corruption robustness. These include i) shape bias of the network, ii) styles that are transferred from paintings and iii) statistics of natural images from IN. The *role of shape bias* is studied using the Edge dataset (E) proposed in Section 3. Other variants study the role of the remaining factors and are explained below:

*Role of stylization* We create Stylized Edges (SE, see Figure 2) for which the styles from the paintings are transferred to the edge maps of Edge ImageNet20 (E). Here, we study the significance of stylization without the presence of the statistics (texture details) of natural images.

*Role of out-of-distribution styles* SIN is generated by transferring the styles from out-of-distribution images, namely paintings. We create its variant called Intra-Stylized IN (I-SIN, see Figure 2) for which in-distribution images from IN are chosen randomly to transfer the styles. We also generate Intra-Stylized Edges (I-SE) where the image styles of IN are transferred to the Edge dataset E.

*Role of natural images statistics* The above variants of E or SE test the role of shape and stylization without retaining texture cues of natural images. We create another variant called *Superposition* (SE+IN, see Figure 2) that interpolates images $I_{SE}$ from SE with images $I_{IN}$ from IN to embed the statistics (texture details) from natural images: $I_{SE+IN} := (1 - \alpha) \cdot I_{SE} + \alpha \cdot I_{IN}$. We set $\alpha = 0.5$.

These different stylized variants including E allow insights into the interplay between shape bias and corruption robustness. For simplicity, we term the networks that are trained on a certain dataset using the name of that dataset. For example, network trained on Stylized Edge (SE) is referred to as SE. The evaluation of SIN and I-SIN reveals the significance of the choice of styles and evaluation of Edge (E) indicates the role of the shape bias for corruption robustness. SE explains the importance of stylization and finally SE+IN allows to understand the importance of natural image statistics that are preserved in SIN and I-SIN but are missing from SE. Table 3 provides an overview of the input image compositions of different variants that are described above.

| Network | shuffled image patches $4 \times 4$ acc(%) | | | shape based cue conflict #400 | | |
|---|---|---|---|---|---|---|
| | No styling | style blending | style randomization | No styling | style blending | style randomization |
| IN | 67.22 | 51.34 | 41.97 | 63 | 82 | 86 |
| SIN | 38.46 | 36.96 | 34.95 | 144 | 155 | 156 |
| E | 34.11 | 33.95 | **28.43** | 155 | 166 | **193** |

Table 1: Comparison of different feature space style augmentation methods on $4 \times 4$ shuffled image patches and number of shape based predictions in texture-shape cue conflict images. Evaluation of shuffled patches is conducted on $598$ correctly classified validation images by all the networks.

**Network details** We employ a ResNet18 architecture with group normalization (Wu & He, 2018) and weight standardization (Qiao et al., 2019). We include *SR* described in Section 3 in the architecture. ResNet18 contains 4 stages of series of residual blocks and *SR* is inserted before every stage. We train ResNet18 on different datasets and their variants described above. IN and SIN are considered as baselines. We show that E possesses more global shape details of the objects whereas SIN demonstrates little or no texture bias for decision making. Both these datasets are complementary to each other and further enhance shape-based predictions when combined (termed as E-SIN). Note that *SR* is used to reduce texture bias and IN contains by far the strongest texture cues. Hence, *SR* is applied only on the training samples of IN but not on other dataset variants. Nevertheless, *SR* applied on other dataset variants found no differences in the results.

**Training details** Network on differnt dataset variants except IN are trained in two stages. The first stage begins with training the network on the respective dataset variant (e.g: E) for a total of 75 epochs starting with a learning rate of $0.1$, which is dropped at the 60th epoch by a factor 10. In the second stage, the networks are then fine-tuned on the respective dataset along with IN (e.g: E & IN) for another round of 75 epochs starting with a learning rate of $0.01$, later reduced to $0.001$ at the 60th epoch. On the other hand, the network on IN is trained for 100 epochs with a learning rate of $0.1$, reduced to $0.01$ and $0.001$ at the 60th and 90th epochs, respectively. We use a batch size of $128$ samples with the SGD optimizer and weight decay $10^{-4}$.

During the fine-tuning stage, we freeze the first convolutional layer and the first normalization layer's affine parameters. We observed that freezing these two layers demonstrate more global shape bias than fine-tuning all the layers in the network. During fine-tuning, the networks receive an equal number of training samples from both datasets (e.g: 128 samples from E and 128 samples from IN in a mini-batch). Note that the data distribution of edge maps from the datasets E, SE and I-SE are different than the distribution of images from other datasets. Fine-tuning the network on inputs with different distributions results in degradation of the performance. In other words, the datasets E, SE and I-SE do not preserve natural image statistics and degrade task performance when finetuning along with clean images. Hence, we weigh the loss of training samples of edge maps from E, SE and I-SE when fine-tuning along with IN. The loss between training samples is weighted as follows: $Loss\ L = L_{\text{IN}} + \lambda L_{\text{edgemaps}}$, with $\lambda = 0.01$. Finetuning on style variants SIN, I-SIN that better preserve natural image statistics does not affect classification performance significantly, hence $\lambda$ is not used. Larger $\lambda$ preserves the shape bias but affects the clean accuracy while smaller $\lambda$ reduces shape bias of the network. In case of E-SIN, We fine-tune the network that is pre-trained on E in the first stage of training with SIN and IN in the second stage and show that such setup further improves shape-based predictions. All ResNet18 models have validation accuracy of about 87% on IN.

## 5 EVALUATION OF SHAPE BIAS

In this section, we evaluate different methods in terms of their shape bias using two different evaluation criteria - *Shuffled image patches* and *Texture-shape cue conflict* that are described below.

**Shuffled image patches:** Following Luo et al. (2019), we manipulate images by perturbing the shape details while preserving the local texture of the objects. We divide an image into different patches of size $n \times n$ with $n \in \{2, 4, 8\}$ and randomly shuffle the patches as shown in Figure

| Network | shuffled image patches acc(%) | | | texture-shape cue conflict results | | |
|---|---|---|---|---|---|---|
| | $2 \times 2$ | $4 \times 4$ | $8 \times 8$ | shape #400 | shape #100 | texture #100 |
| IN | 78.57 | 41.93 | 31.21 | 86 | 18 | 20 |
| SIN | 75.78 | 35.56 | 18.48 | 156 | 32 | **2** |
| E | 73.29 | 28.42 | 11.18 | 193 | 46 | 15 |
| SE | **66.77** | 28.73 | 12.89 | 224 | 55 | 6 |
| E-SIN | 71.12 | **23.76** | **10.25** | **234** | **58** | 6 |

Table 2: Comparison of models trained on different datasets on shuffled image patches and number of texture-shape cue conflict predictions based on shape and texture labels. Evaluation of shuffled image patches is conducted on 644 validation images that are correctly classified by all the networks.

**A2a.** Larger $n$ corresponds to more distorted shapes. The performance of networks that rely more on shape is expected to deteriorate more strongly as the number of patches increases. We conduct this evaluation only on the ImageNet20 validation images that were correctly classified by *all* the networks that are selected for comparison.

**Texture-shape cue conflict:** The cue conflict image dataset proposed by Geirhos et al. (2019) consists of images where the shape of an object carries the texture of a different object. For example, the object `cat` holds the texture of `elephant` as shown in Figure A2b. Each image in the dataset contains two class labels: labels with respect to shape and texture. The evaluation is carried out to test the network's bias towards shape or texture. Networks with strong shape bias will exhibit higher accuracy according to the shape label while networks with texture bias will have higher accuracy for texture-based label. The original dataset contains a total of 1280 cue conflict images designed for the evaluation of the networks trained on the entire ImageNet dataset. 400 of these images have classes (shape labels) present in ImageNet20. A subset of 100 instances (20 instances from 5 different categories) from the selected images also has a texture label that belongs to ImageNet20 (see Figure A2b bottom). The remaining 300 images with texture labels that do not belong to the classes of ImageNet20 are not considered for texture-based classification.

**Results** The results in Table 1 compare *style blending (SB)* (Nam et al., 2019), *style randomization (SR)* (Section 3), and no styling in feature space for networks trained on IN, SIN and E. In terms of performance on $4 \times 4$ shuffled patches, *SB* performs worse than no styling, and *SR* performs even worse than *SB*. This indicates increasing shape bias from no styling over *SB* to *SR*. This finding is reinforced by an increasing number of images classified according to the shape label for texture-shape cue conflict images from no styling over *SB* to *SR*. Similarly, when comparing different training datasets, SIN results in stronger shape bias than IN, and E exhibits stronger shape bias than SIN.

In Table 2, we compare additional networks, all with *SR* enabled. Here, we again see a consistent trend of increasing shape bias from IN over SIN to E. Moreover, stylized edges (SE) further increase shape bias than E. Lastly, E-SIN improves shape bias even slightly beyond SE. In summary, we can see a clear increase in shape bias for the methods proposed in this paper over IN or SIN. Next, we investigate to which extent this also results in an increased corruption robustness.

## 6 INFLUENCE OF SHAPE BIAS ON COMMON CORRUPTIONS

We compare different networks in terms of their corruption robustness. Figure 3 shows the accuracy of different networks for two types of corruptions: Gaussian noise and frost (refer Figure A5 for all corruptions). Table 3 presents the corruption accuracy averaged over 15 ImageNet-C corruptions along with shape and texture results on the texture-shape cue conflict dataset. Generally, a CNN trained on IN performs poorly in terms of corruption robustness while SIN is relatively robust. On the other hand, E performs considerably worse than SIN and is not consistently better than IN despite having an even stronger shape bias than SIN. Networks SE and E-SIN further increase shape bias but

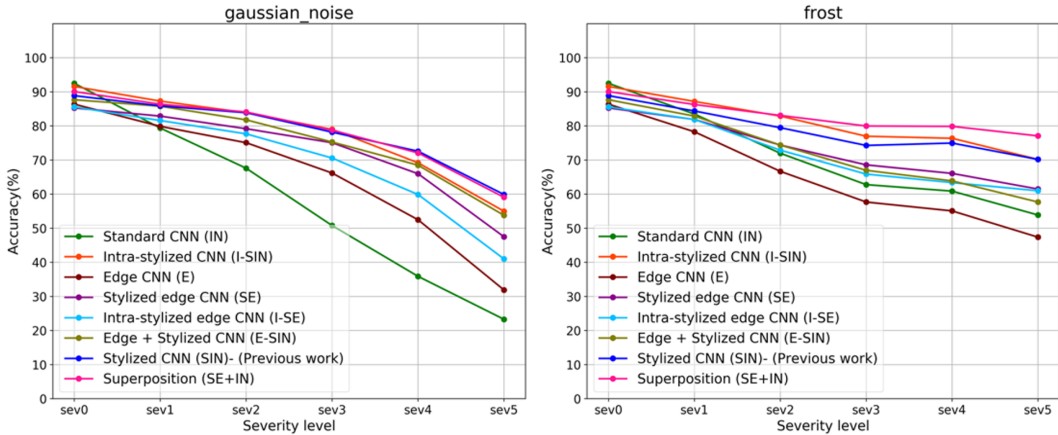

Figure 3: Classification accuracy of different networks on two corruptions across 5 severity levels. Severity 0 represents accuracy on clean validation data of IN. Severity levels 1 - 5 follow the corruption parameters from Hendrycks & Dietterich (2019) and represent increasingly strong corruptions.

| Network | Input image composition | | | Shape #100 | Texture #100 | Mean corruption acc(%) |
|---|---|---|---|---|---|---|
| | Natural image | Edge map | Style transfer | | | |
| IN | ✓ | ✗ | ✗ | 11 | 39 | 64.69 |
| SIN | ✓ | ✗ | ✓ | 34 | 2 | 77.64 |
| E | ✗ | ✓ | ✗ | 46 | 15 | 62.01 |
| SE | ✗ | ✓ | ✓ | 55 | 6 | 71.81 |
| E-SIN | ✓ | ✓ | ✓ | 62 | 5 | 71.55 |
| SE+IN | ✓ | ✓ | ✓ | 22 | 13 | **78.96** |

Table 3: Mean corruption accuracy (mCA) and texture/shape results on texture-shape cue conflict dataset of different networks. mCA is the mean accuracy over 15 ImageNet-C corruption and severities ranging from 1 to 5. Networks trained with style transfer augmentation perform better than those without and network trained on superpositioned images (SE+IN) yield best mCA.

still perform considerably worse than SIN in terms of corruption robustness. These results contradict the hypothesis that stronger shape bias results in increased corruption robustness.

The only method that slightly surpasses SIN in terms of corruption robustness is the superposition of SE with natural images (SE+IN). However, this method has a relatively small shape bias. A common theme of SIN and SE+IN is that both exhibit properties of a natural image but are strongly distorted by stylization (see Figure 2). We hypothesize that these methods correspond to strong augmentation methods that stay close enough to the data manifold while inducing high diversity in appearance and thereby encourage learning robust representations, which need not necessarily be shape-based. We extend these findings to larger datasets with 200 classes of ImageNet, deeper architectures like ResNet50, DenseNet121, MobileNetV2 and different normalization layers like BatchNorm in Section A.5. Lastly, as can be seen from Figure 3, intra-stylization is nearly as effective as stylization based on paintings, implying that style need not necessarily be out-of-distribution for being useful.

## 7    ON THE ADAPTABILITY OF LEARNED REPRESENTATIONS

As seen in the previous section, style augmentation on natural images is important for the network to be able to generalize to different domains such as common corruptions. We now study how easily

| Network | Corruptions acc(%) | | | | SIN val acc(%) | Cue Conflict | |
|---|---|---|---|---|---|---|---|
| | **Speckle noise** | **Gaussian blur** | **Frost** | **Pixelate** | | **shape #400** | **texture #100** |
| IN | 61.28 | 42.96 | 66.62 | 78.54 | 42.0 | 63 | 39 |
| IN (fine-tuned) | 82.7 | 77.3 | 81.02 | 87.02 | 68.0 | 130 | 13 |
| E | 67.76 | 44.48 | 61.04 | 70.94 | 62.3 | 193 | 15 |
| E (fine-tuned) | 80.18 | 71.74 | 73.7 | 74.78 | 72.4 | 222 | 9 |

Table 4: Mean corruption accuracy, SIN and cue conflict results of networks with & without additional fine-tuning of the affine parameters of normalization layers on the respective corruptions. Fine-tuned networks perform significantly better, despite only the normalization layers are updated.

a pre-trained network can be adapted to a different distribution such as corruptions. Importantly, this uses the "unknown" distortion during training; this experiment is not meant as a practical procedure for the ImageNet-C benchmark but rather for understanding internal representations of a network.

Chang et al. (2019) showed that domain-specific affine parameters in normalization layers are essential when training a network on different input data distributions jointly. We conduct a similar experiment with the key difference that our network is already pre-trained on IN/E and only the affine parameters of normalization layers are fine-tuned to fit the distribution of the respective target domain. First, we fine-tune affine parameters of the network on several ImageNet-C corruptions separately and evaluate the mean corruption accuracy on the same corruption across different severity levels. As shown in Table 4 (left), performance on the corruptions can be greatly improved even with fixed convolutional parameters trained on IN/E by just tuning the affine parameters. Similarly, we also fine-tune the affine parameters of pre-trained CNN on SIN. Results in Table 4 (right) show not only an improvement on SIN validation accuracy but also improved shape-based classification results on texture-shape cue conflict images. These results suggest that the standard CNN encodes robust representations that can be leveraged when adapting affine parameters on a target domain.

## 8 CONCLUSION

We performed a systematic empirical evaluation of the hypothesis that enhanced shape bias of a neural network is predictive for increased corruption robustness. Our evidence suggests that this is not the case and increased shape bias is mostly an unrelated byproduct. Increased corruption robustness by image stylization is better explained as a strong form of augmentation which encourages robust representations regardless whether those are shape-based or based on other cues. We conclude that if corruption robustness is the main objective, one should not primarily focus on increasing the shape bias of learned representations. Potential future research directions will focus on understanding whether shape-biased representations offer advantages in other domains than corruption robustness (Hendrycks et al., 2020). Moreover, one could try devising stronger augmentation procedures in image or feature space based on our findings. Lastly, gaining a better understanding of which kind of features (if not shape-based ones) are essential for corruption robustness is an important direction.

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
