# OpenReview forum: "Does enhanced shape bias improve neural network robustness to common corruptions?"
_ICLR.cc/2021/Conference — ICLR 2021 Poster_

### Official Review · AnonReviewer1 · 2020-10-25
**Take a step further on understanding shape-based representation**

**Rating:** 6
**Confidence:** 4

**Review:**

This paper delves deeper into understanding shape-based representation of CNNs in an empirical way. Based on the stylized images, it proposes to use edge maps to more explicitly feed shape information to learning models. Besides, the common way to let models learn the shape-based representation is to train on the dataset contained the shape information while the texture information is severely distorted. The paper takes the point of changing the statistics of feature maps would result in style changes and proposed style-randomization to help CNNs better focus on shape information. Also, it connects the biasing degree on shape information of models with the defensive performance against common corruptions, like Gaussian additive noise, blur, and etc. An intuitive conclusion was drawn that there is no clear correlation between shape bias and robustness against common corruptions, and justified by extensive experiments.

The studied direction is important for understanding the learned representation of CNNs. The proposed points, using edge maps and style randomization, are simple and effective for making CNNs focus on shape information. Also, the discussion with the common corruptions is intriguing and helps us better understand the relationships between the shape-based representation and the robustness.

The experiments are technically sound and results are sufficient to support their arguments, though more analysis would add support to the claims. The experimental setups are described in detail that makes it easy to replicate the experiments. Overall this paper is easy to follow. The reviewer would encourage the author to share their codes and datasets.

-------------------------------------------------------------------
Some concerns and comments are listed below:

Since most of the experiments are conducted on ImageNet-20 and ResNet-18 (appendix lists few results on ImageNet-200 and ResNet-50), the reviewer doubts if the conclusions are still valid on different datasets and architectures. Especially for the aspects of the architecture, different architectures have different inductive biases and may result in different phenomena.

According to Table 2, 3 and discussion on the bottom of the 6th page, the Stylized-Edge data would increase shape bias larger than Edge. This is a little bit counter-intuitive due to Stylized-edge only contain more uncorrelated texture information. The reviewer wonders if the quality of the Edge data is not very high and result in the phenomenon. To verify this, the author better to do experiments on the high-quality edge data with human annotations or performing edge algs on simple data.

Table 3 and A5 list the results of E-SIN and SE+IN. The reviewer wonders if there are any "intermediate" results such as E-IN, E+IN to better compare. Also, the reviewer is curious about the useful of superposition. Will the concat or mix data bring similar benefits, compared to superposition?

Currently, to test the degree of focusing on shape information, the author runs the random-shuffled images and tests on stylized-imagenet. This is good but the reviewer wonders if we directly test on the edge maps of validation sets.

To sum up, the reviewer thinks this paper would bring some new understandings to the community.

--------------after rebuttal-------------------

I've read all reviews and the rebuttal, and thank the authors for their efforts and extra experiments. I think it is ok to be accepted due to provide further understandings about the learning representation with solid experimental results.

---

> ### Author Response · Authors · 2020-11-25
> **Thank you for raising interesting questions. The additional results support our findings.**
>
> We thank the reviewer for the helpful review & raising interesting questions that provide more insights in our work. We address reviewer's key points below:
>
> 1) Do the conclusions from this paper valid on different datasets & architectures ?
>
> Conclusion on different architectures:
> Thanks for raising this point that adds value and improve our work. We extended our findings from ResNet to other architectures like DenseNet121 & MobileNetV2 and the results on these two architectures can be found in Table A6 & Table A7 under Section A.6.5 in the revised supplementary material. Similar to the results on ResNet, we find that E-SIN exhibits a higher shape bias than other settings but still has a lower mCA than SIN. On the other hand, Superposition(SE+IN) shows a lower shape bias with similar mCA to SIN. These results show that our conclusion is also valid using different neural architectures.
>
> Conclusion on different datasets:
> We agree with the reviewer that extending our findings to different datasets (e.g. CIFAR10 & CIFAR100) would improve significance of our work.
> Three main reasons why we focus our experiments on subsets of ImageNet as a dataset are:
>
> (i) the influence of stylized data on corruption robustness and the hypothesis related to the shape bias has been investigated on ImageNet dataset originally (Geirhos et al. (2019)),
>
> (ii) the availability of state-of-the-art style transfer techniques and high quality edge detection algorithms on larger resolution images,
>
> (iii) availability of texture-shape cue conflict images for shape based evaluation.
>
> However, results using ImageNet20 (a closer analogy to CIFAR-10 with respect to smaller number of classes) and also from larger dataset with 200 classes - ImageNet200 (please refer Section A.6.2 in the supplementary material) have shown that our findings are not limited but valid on both other datasets. We recall our finding - "corruption robustness benefit from style variation in the vicinity of the image manifold, but not the shape bias and interpret the image stylization as a strong data augmentation technique". We believe that this conclusion is also valid on different datasets based on on the improved corruption robustness observed in AugMix (Hendrycks* et al., 2020), which shows that an effective data augmentation strategy in the vicinity of image manifold improves network robustness under data shift.
>
> 2) Stylized-Edge data increase shape bias larger than Edge.
>
> This is indeed an interesting observation, and we suppose it is due to the following explanation: stylization introduces texture to the edges and thereby brings the edge dataset closer to the distribution of natural images and texture-shape cue conflict dataset. The shape details in the form of edges along with the matching distribution makes the Stylized-Edge improve shape bias more than Edge.
>
> 3) Results on intermediate settings E-IN, E+IN to better compare.
>
> Thanks for pointing out these interesting settings. We compare these two settings in the results of DenseNet121 and MobileNetV2 in Table A6 and Table A7 under Section A.6.5 in the supplementary material. We notice that E-IN exhibits a similar shape bias and corruption results as setting E, whereas E+IN possess lower shape bias but serves as a simple data augmentation strategy (similar to superposition SE+IN) that improves corruption robustness over IN.
>
> 4) Will the concat or mix data bring similar benefits, compared to superposition ?
>
> It was shown in the previous work AugMix (Hendrycks* et al., 2020, Figure 5 of their work) that mixing the data with the corresponding augmented version is beneficial compared to just mixing different data points (Mixup). In a similar manner, we designed Superposition such that it mixes images with their stylized variants rather than mixing different data points. Exploring alternatives to superposition would be an interesting direction for future work. We thank the reviewer for this interesting suggestion.
>
> 5)Test the degree of focus on shape information using edge maps of validation sets.
>
> We perform shape-based evaluation using random-shuffled images and texture-shape cue conflict images. Among the different settings in our experiments, edge maps of training data are directly used in E throughout the training, also used for pretraining the network in E-SIN, and stylized edge maps of training data are used during training of Superposition (SE+IN). On the other hand, edge maps are not used in any way for training in IN and SIN (please refer Section A.3 for training details. We will move these training details to the main paper in our final version for better understanding of the training settings).  Hence, the evaluation of shape bias based on the edge maps of validation data would favor E, E-SIN and SE+IN and would not provide a fair comparison. However, we provide the performance of different settings on edge maps of validation set in Table A8 under Section A.6.6 in the revised supplementary material.

---

### Official Review · AnonReviewer2 · 2020-10-28
**Clear, thorough and convincing demonstration that shape-bias does not inherently confer corruption robustness**

**Rating:** 9
**Confidence:** 5

**Review:**

##Updated Review##

I'd like to thank the reviewers for their responses, and for updating to the much clearer naming scheme for the different methodologies!  Much easier to follow with those names.

I maintain that this is high quality novel work that contradicts a widely held belief within the field and as such is a clear accept.

# Main Idea
The main idea is that training to reduce texture bias in convnets in favor of shape bias is often thought to cause the concomitant increase in corruption robustness, but this has not been tested directly.  The authors propose a systemic study of this relationship and conclude that both shape bias and increase corruption robustness are both byproducts of style-variation during training, that is they share a common causal mechanism, but that shape bias does not itself cause corruption robustness.

They accomplish this by creating a new augmented dataset which encourages learning shape features (created from the edge maps of the training images) but does not also induce robustness against common corruptions.

#### Additional findings from the study:
Shape bias gets maximized when edge information and stylization are combined without including any texture information.

Corruption robustness is maximized by superimposing the image (and its textures) on the above stylized edges.

They propose that corruption robustness seems to benefit most from style variation in the vicinity of the image manifold.

# General Strengths
I think the paper makes a compelling case.

The Geirhos result, while elegant and interesting, has always struck me as a roundabout way to get to the shape bias question.  These authors more directly attack it, and show that the relationship doesn’t hold, while showing the real. Value of the Geirhos result was in showing that manifold-local variation increased robustness (and also consequently shape bias).

In addition to the elegance of the result, the authors use of “texture randomization via texture feature randomization” is a an elegant (and much faster) implementation than regenerating an entire dataset with many different textures.   The insight is good, the image itself doesn’t need to be rerendered, but the networks interpretation of its texture needs to be scrambled.  This doesn’t restrict you to the statistics of any database on which you might extract textures

# Weaknesses
Doesn’t report I-SIN results across ImageNetC corruption dataset.
Would have preferred to see the specific corruption type results broken out for more than just the two shown and the average robustness (the variation and performance on different types of corruption could be useful information).

Also make sure the names are consistent across graphs and paragraphs.  Not the best naming scheme (too many very similar acronyms which aren’t immediately evocative of the underlying point).

---

> ### Author Response · Authors · 2020-11-25
> **Thank you for the precious feedback!**
>
> We thank the reviewer for providing invaluable feedback and raising points that improve readability of our work.
>
> 1) I-SIN results across ImageNetC corruption dataset
>
> The mean corruption accuracy of I-SIN is 77.8\% which is similar to SIN 77.64\% in Table 3. Thanks for pointing out this missing detail. We will include this result in our final version.
>
> 2) Performance on each individual corruption
>
> We would like to point to Figure A5 in the supplementary material for the results on different corruptions of ImageNet-C at five different severity levels.
>
> 3) Choose better naming schemes
>
> As suggested by the reviewer, we will update the naming schemes of different training settings and make them consistent across graphs and paragraphs for better readability in our final version. We will replace the current naming scheme as follows:
>
> IN         - Standard
>
> SIN       - Stylized
>
> I-SIN    - IntraStylized
>
> E           - Edge
>
> SE         - StylizedEdge
>
> I-SE      - IntraStylizedEdge
>
> E-SIN   - Hybrid
>
> SE+IN  - Superposition

---

### Official Review · AnonReviewer3 · 2020-10-29
**I vote for ‘marginally above acceptance threshold’. While the novelty of the paper seems to be insufficient and the experiments seems to have several weaknesses, the paper presents an interesting analysis towards the relationship between shape bias and robustness to corruptions of neural networks**

**Rating:** 7
**Confidence:** 4

**Review:**

The paper disproves the hypothesis that addressing shape bias improves robustness to corruptions of neural networks, which has been stated by the previous studies [1, 2]. The paper demonstrates that the degree of shape bias of a model is not correlated with classification accuracy on corrupted images via experiments. For the experiments, this paper presents two novel methods to encourage CNNs to be shape-biased: 1) edge dataset and 2) style randomization (SR). In the experiments, the authors train CNNs to be shape-biased to various degrees based on the proposed methods. Additionally, they compare test accuracies of the models to evaluate shape bias and robustness to corruption. In addition, this paper shows that through fine-tuning the affine parameters of the normalization layers, a CNN trained on original images can achieve comparable, if not better, performance than a CNN trained with data augmentation.

Pros:

- The paper clearly demonstrates that shape bias is not correlated with robustness to corrupted images. The test accuracy on texture-shape cue conflict images effectively indicates the degree of a model’s shape bias. Also, the authors visualize the results that explicitly compare the models’ accuracy on corrupted images.

- The authors present an interesting analysis towards Stylized ImageNet (SIN) dataset by separating the dataset into different factors that are used to generate SIN. According to this paper, the newly made datasets based on each factor leads a CNN to be shape-biased to various degrees. This provides insightful perspectives on shape bias and corruption robustness. Specifically, it is intriguing that a model trained on a stylized dataset using styles from in-distribution images achieves comparable performance compared to what uses styles from out-of-distribution images, such as paintings.

Cons:

- The novelty of the two proposed methods, edge dataset and SR, is limited. First, the edge dataset is created based on an existing model. The authors simply convert the non-binary edge map into a binary one. Furthermore, there are not enough experiments or evidence that validate the effectiveness of this method compared to the original method. Additionally, SR is also similar to existing methods, except that they change the target distribution from training samples’ distribution to a uniform distribution.

- According to this paper, a CNN trained on edge dataset is more shape-biased but achieves lower accuracy on corrupted images compared to a CNN trained on SIN. The authors state that the result indicates that shape bias is not correlated to corruption robustness.
However, it seems to be unfair to compare edge dataset and SIN to disprove the correlation between shape bias and corruption robustness.
Edge dataset, unlike SIN, does not contain any information except edges. Therefore, a CNN trained on edge dataset ‘learns only shape information, while a SIN-trained CNN learns other information as well, but ‘focuses more on the shape’. Since the edge dataset provides much less information of the original images, it is trivial that a CNN trained on edge dataset achieves lower performance compared to a SIN-trained CNN.
The previous approaches [1, 2] mentioned in this paper also imply that shape bias is basically related to how much the model focuses on shape information, not to how much shape information the model is given.
Therefore, it would be more reasonable to compare a SIN-trained CNN with other CNNs that are trained on datasets containing a similar amount of information, but have different levels of concentration on shape information.

[1] Geirhos et al., “ImageNet-trained CNNs are biased towards texture; increasing shape bias improves accuracy and robustness.” ICLR’19

[2] Michaelis et al., “Benchmarking Robustness in Object Detection: Autonomous Driving when Winter is Coming.” ICLR’20


--------------------------
After rebuttal:

Thank you for the additional explanation.

The comments by the authors effectively address my concerns. Although the edge dataset and SR are quite similar to the existing methods, the authors clearly present the usefulness of them as well as their additional contributions. Also, additional training details support the adequacy of the comparison in the experiments. Therefore, I would like to increase my final score to 7: Good paper, accept.

---

> ### Author Response · Authors · 2020-11-25
> **Thank you for providing thoughtful feedback**
>
> We thank the reviewer for providing valuable feedback that helps us to clarify some of the key points in our work.
>
> 1) Novelty of the two proposed method is limited.
>
> We agree with the reviewer that the edge dataset is created based on existing pretrained model. However, similar to Geirhos et al. (2019) that proposed to train on stylized data, we propose to train on the edge dataset to encourage learning shape-based representations. Although Style Randomization (SR) is similar to an existing method, the differences make SR superior in the context of shape-based evaluation. Our contribution in this work is not limited to the proposal of these two methods for higher shape bias but also includes an analysis of the role of shape-based representations on corruption robustness, falsifying the hypothesis established in previous work (Geirhos et al., 2019), and providing a clear understanding of the properties of the training data that improve corruption robustness.
>
> 2)  Unfair to compare edge dataset and SIN to disprove the correlation between shape bias and corruption robustness.
>
> Thanks for raising this point. Note that both the network settings Edge (E) and Stylized ImageNet (SIN) also receive the original images of ImageNet (IN) as input during training (please refer Section A.3 for training details). We will move these training details from the supplementary material to the main paper in our final version for a better understanding of the training settings. We agree with the reviewer that the edge dataset contains less information on the original images. However, we train the model with edge dataset along with original images and show that the model that is provided with shape details in the form of edge maps improves its focus on global shape information (please refer Table.2 for the shuffled image patches evaluation results). In addition to the edge dataset, we compare SIN with other dataset variants like Stylized Edges (SE) and also other training scheme E-SIN. Both these settings possess a higher shape bias but show lower corruption robustness than SIN. Lastly, a model trained on Superposition (SE+IN) that contains information on the original images and is augmented with different styles has a lower shape bias but shows higher corruption robustness than E, SE, E-SIN and is comparable to SIN. We use these empirical evidences to disprove the correlation between shape bias and corruption robustness.

---

### Official Review · AnonReviewer4 · 2020-11-02
**interesting results and experiments, might need more comprehensive experiments**

**Rating:** 6
**Confidence:** 2

**Review:**

Summary of the paper:
This paper tries to study whether increasing shape-bias of a neural network trained with imagined will make it more robust to common corruptions such as gaussian noises.
The paper falsified this point by producing a data augmentation method which leads to more shape biased network yet less susceptive to common corruptios.
The paper further hypothesize that it’s the stylization augmentation that leads to increase robustness of the network by ablation studies.

Strength:
The paper does provide a counter example to the common hypothesis that increase shape bias can lead to more robust network. This provides insight to future researches on understanding how shape-bias and texture-bias affects on neural network robustness. If the results of the counter example is reproducible and significance, then the claim is convincing and the paper did a good job verifying such hypothesis.

Weakness:
1. The experiment results seemed to be limited in this dataset. In order to make such general claim, I would expect results for different datasets (e.g. CIFAR), large scale datasets (e.g. the whole imagined), and different network and training procedure (e.g. not just resent).
2. This might be some minor things, but it would be nice if there are statistic significance test for the results (or at least show the variance of couple runs). When I looked at the difference of the number, it would be nice that one can make sure such results is statistically significant with respect with difference runs.
3. Most of the paper is very empirical, and there is little insight or theory or principal ways that organize the results.


Justifications:
While I do like the paper that provide insight and show negative results falsifying some prevalent claim, but the paper provides rather limited evaluation without theoretical insights. Since I’m not an expert in this field, I will recommend borderline scores to hear about the authors’ response.

-------------------
UPdate: the authors' reply address my concerns well, so I raise my rating to the acceptance side.

---

> ### Author Response · Authors · 2020-11-25
> **Thank you for the valuable feedback**
>
> We  would like to thank the reviewer for raising key points and help us to clarify the contribution in our work.
>
> 1) Experiment results for different network architectures and datasets
>
> Extension to different architectures:
> Thanks for suggesting us to extend our results to different architectures that help us to improve our work. We extended our findings from ResNet to other architectures like DenseNet121 and MobileNetV2 and the results on these two architectures can be found in Table A6 and Table A7 under Section A.6.5 in the revised supplementary material. Note that these results also include additional network settings E-IN and E+IN that are suggested by Reviewer1. Similar to the results on ResNet, we find that E-SIN exhibits a higher shape bias than other settings but still has a lower mCA than SIN. On the other hand, Superposition (SE+IN) shows a lower shape bias with similar mCA to SIN. These results show that our conclusion is also valid using different neural architectures.
>
> Comment on different datasets:
> We agree with the reviewer that extending our findings to different datasets (e.g. CIFAR10 and CIFAR100) would improve significance of our work.
> Three main reasons why we focus our experiments on subsets of ImageNet as a dataset are:
>
> (i) the influence of stylized data on corruption robustness and the hypothesis related to the shape bias has been investigated on ImageNet dataset originally (Geirhos et al. (2019),
>
> (ii) the availability of state-of-the-art style transfer techniques and high quality edge detection methods on larger resolution images,
>
> (iii) the availability of texture-shape cue conflict images for shape based evaluation.
>
> However, results using ImageNet20 (a closer analogy to CIFAR-10 with respect to the smaller number of classes) and also from a larger dataset with 200 classes - ImageNet200 (please refer Section A.6.2 in the supplementary material) have shown that our findings are not limited to ImageNet but valid on both other datasets. We recall our finding - "corruption robustness benefit from style variation in the vicinity of the image manifold, but not the shape bias and interpret the image stylization as a strong data augmentation technique". We believe that this conclusion is also valid on different datasets based on the improved corruption robustness observed in AugMix (Hendrycks* et al., 2020), which shows that an effective data augmentation strategy in the vicinity of the image manifold improves network robustness under data shift.
>
> 2) Statistic significance test for the results
>
> From our initial experiments, we observed that our results are consistent across different runs. As suggested by the reviewer, we will add error bars in our final version.
>
> 3) Very empirical, and provides rather limited evaluation without theoretical insights.
>
> We agree with the reviewer that our work is more empirical. However, our work presents a detailed and systematic study to falsify the hypothesis established in previous work (Geirhos et al., 2019), analyzes the role of shape-based representations on corruption robustness, and provides a clear understanding on the properties of the training data that improve corruption robustness. We believe that our findings will steer the direction of future works on corruption robustness by shifting the primary focus from increasing the shape bias to devising stronger data augmentation strategies.

---

### Public Comment · ~Dinghuai_Zhang1 · 2020-11-11
**Related work**

Congrats on your reviews and I really enjoy reading your submission. I'm writing the comment to introduce our *HIGHLY* related ICML work:

Shi B. et al, Informative Dropout for Robust Representation Learning: A Shape-bias Perspective, ICML2020. https://arxiv.org/abs/2008.04254

---

> ### Author Response · Authors · 2020-11-25
> **Thanks for the relevant reference**
>
> Thanks for pointing us to this highly relevant work. We will add a reference to it to the final version of the paper.

---

### Author Response · Authors · 2020-11-25
**We thank all the reviewers for providing valuable feedback**

We greatly appreciate that you consider our work clear [R2], convincing [R2, R4], intriguing [R1, R3] and provides insights for better understanding of the relation between shape bias and corruption robustness [R1, R3, R4]. As suggested, we have extended our results using different architectures like DenseNet121, MobileNetV2 and updated them in the supplementary material. Below we provide detailed responses to your main criticism and helpful suggestions. We will update our paper accordingly for the final version with the discussion involved in this forum, which include clarifications to the key points raised by reviewers, report the mean corruption accuracy on I-SIN, add reference to Shi B. et al as pointed by Dinghuai Zhang. Currently, we term the networks that are trained on a certain dataset variant using the name of that dataset for simplification. In our final version, we will term these two differently and also modify the naming schemes of different training settings for better readability.

---

### Decision · Program_Chairs · 2021-01-07
**Final Decision**

**Decision:**

Accept (Poster)

**Comment:**

This work investigates the recently proposed hypothesis that enhanced shape bias improves neural network robustness to common corruptions. Several interesting experiments are performed to better understand the contributing factors that lead to improved robustness of models trained with texture randomization. Of particular note, the authors design a data augmentation strategy that verifiably increases the shape bias of model, but for which corruption robustness is not improved. Reviewers agreed that this is an interesting counter-example to the shape-bias hypothesis and improves our understanding of why stylization improves robustness. Given the carefully designed experiments investigating an important topic I recommend accept.